# Position: Explainable AI Is Misleading Paradigm Shifting of Neural Network

## Abstract

The position paper argues that the neural network is shaping the new paradigm of science, and neural network is generally superior to the rule, logic, and mathematics. The network hypothesis emphasizes the superiority of the neural network: (I) The neural network is not interpretable in the arbitrary combination of linear models; (II) There is one general form of the common state with the accuracy in efficiency for any neural network. The new paradigm is potentially shaping the new promising area to unlock the unknown fundamental mechanism behind the neural network, and the obtained representation is thoroughly distinct with the expectation in the conventional approximating mathematics.

## 1. Introduction

The high accuracy and extraordinary emergence suggest the superiority of the neural network in efficiency to simulate the real scenarios in the performance in the competition with the other algorithms and backend methods. The neural network achieves the magnificent triumphs with the scientific discovery and technological breakthrough in the cross-domain applications, but its specialty and uniqueness remain undervalued in the couple of the mysterious phenomenon around the engineering practices since the recent renaissance of the deep learning in millennium. Training with the supervised criterion naturally leads to the representation in the hidden layer in the neural network (Goodfellow et al., 2016), and the positive relationships are sometimes ambiguous in the strict manner between the variables and the responses within the tricky data-driven process.

**Neural networks are establishing a new paradigm in the AI research, demonstrating superiority over rule-based systems, logic sequencing, and traditional mathematics approaches.** The novel paradigm is revolutionizing the current science with the superior advantages over rule-driven, logical reasoning, and mathematical methodologies in our knowledge.

**The interpretability hypothesis** (Section 3) formally breaks the linearity assumption on explaining the neural network in the naive simplification. The neural network is not explainable with the combination of the linear models and compound functions, and the linear equation is conversely only the specialized form of the neural network. The neural network could better efficiently elaborate the generality in the scenario analysis beyond the boundary of the finite form of the function in mathematics. The individual neurons are not as the same additive components as in the spelled equations.

**The universality hypothesis** (Section 5) reveals the unique status in selecting the possible states from the model training process. The common state is the universal status with the representation of the complex system with the reasoning ability. The ideal common state is not perfectly identifiable under the current training and reference stages, but the bias and deviation really become the matters from the ideal status for the accuracy in the neural network and language models. The common state is leading the procedures as the universal catalyst avoiding the models into the underfitting states or the overfitting states caused from the benchmark, the metrics and the data. The scale of the parameters facilitate the selection process in abstracting the complex representation to equip with the acquired ability and the safety-related hallucination.

The theoretic pursuit is absent and invalid in explaining the zero-shot performance and few-shot adaption for the neural network and transformer-based model, and we still have no literature unveiling the mystery with the decent paradigm and concrete theory on understanding the black magic and potential threats in explainable Artificial Intelligence. The emergent capability and the hallucination might be the two sides of the large language models. The suddenly acquired ability has the zero-shot fashion without any parameter update in the model (Bai et al., 2023), and the jailbreaking attacks are threatening the society safety with the harmful generation from large language models (Ji et al., 2023). DeepSeek R1 naturally emerges with numerous powerful and intriguing reasoning behaviors with the post-training

[1]Anonymous Institution, Anonymous City, Anonymous Region, Anonymous Country. Correspondence to: Anonymous Author <anon.email@domain.com>.

Preliminary work. Under review by the International Conference on Machine Learning (ICML). Do not distribute.

large scale reinforcement learning in prioritizing helpfulness and harmlessness (DeepSeek-AI et al., 2025).

The neural network hypothesis characterizes the complex possibility and extreme flexibility on neural network from the perspective on end-to-end modeling. **Interpretability hypothesis** (Section 3) emphasizes the inner complexity of neural network in the parameter and weighting, instead of only capturing the interactions in the linear approximation. Barron theorem (Barron, 1993) is the earlier theoretical trial to approximate the defined smooth functions with the invariant form of the neural network, and the followers of the seminal studies extend to avoid the curse of dimensionality with the universal approximation (Bach, 2017; Lawrence, 2022). The corollary (Section 4) is the first step to explain the polysemantic with the assumption on the **neuron flexibility** with the conjecture to abstract the internal structure on neuron activities interacting with the external problems. The multi-functional neurons enable the network flexibility to encode the hierarchical knowledge structures. From the recent pruning attribution graphs with the visualization by Anthropic, an extra pruning step explore the phenomenon that the neurons are typically polysemantic in performing many different functions that are seemingly unrelated (Lindsey et al., 2025). The corollary is the hypothesis on the neural network, and distinct with the recent exploration from the interventionist perspective. **Universality hypothesis** (Section 5) is the ideal presumption on considering the representation with the data feeding in the model training, and the common state has the greater potential to guide the practice fixing the problem of inappropriate drawback and catastrophic forgetting. The **reverse availability** suggests the alternative solution obtainable in unifying the blocks of the small networks to emerge capability with the universal things. The **deficiency hypothesis** (Section 6) assumes that hallucination and modeling shortcomings are inheriting from the internal structure imitating the neural activation and network response in brain. The theoretical foundations still remain invalid in internal process and adaption mechanism for the emergent abilities and hallucinations of the large language models. Some desirable qualities of LLMs are not a consequence of statistical generalization and require the separate explanation in theory (Reizinger et al., 2024).

The progressive urgency and Responsible AI are also boosting the theoretical blocks more substantive on the internal essence and the observed phenomenon without the imagined boundary. The initial expectation is allowing mimicking human-like intelligence for the machine and robots as the empathetic companies, insightful advisors and tireless innovators, but the realization of the embryonic and super human Artificial General Intelligence are potentially too advanced in enabling the systems with the consciousness and self-improvement (Feng et al., 2024). One of the in-

termediate possibilities of the co-creative AI systems is the synthesis of human experiences and machine creativity (Rezwana & Maher, 2024). The developer community is partly in the conservative attitude to prevent the potential threats to the physic world in controlling the open source strategy on modeling parameter and training secrets of the large language models. Responsible AI additionally advocates for the development of systems aligning with ethical values as fairness and transparency (Sadek et al., 2024). The detailed explanations improve the informational and distributive fairness perceptions (Aljuneidi et al., 2024). The interdisciplinary dialogue is necessary between regulatory authorities, legislation professionals and social scientists (Gray et al., 2024).

The proposed network hypothesis is in the earlier phase on shifting the explainable AI into the new paradigm with the interpretability (Section 3) and universality (Section 5). The novel contributions are not limited with the decent explanation in the unparalleled insights on dropout in network, the multiple paths in continuous training procedures, and more meaningful ongoing discussion is positive as following,

- The neural network is powerful and superior in representing the symbolized world with the complex possibility (Section 3) and extreme flexibility (Section 4).

- The representation is more prominent in the common state than the generalized algorithms with the functions and equations (Section 5).

- The representation simultaneously have the emerged capability and unexpected hallucination in imitating human reasoning and perception (Section 6).

- Neural network is the distinct form without the approximation and finite equalization in the contemporary mathematics.

## 2. Literature Review

**Connectionism** renaissance dates back to the earlier foundation in the relational network with the certain grammatical constraints. The neural network has the obvious obstacle in violating of the weak rules efficiently and appropriately (Hinton, 1977). The approximation rate and the parsimony of the parameterization of the neural networks are theoretically advantageous than the high-dimensional settings of the polynomial, spline, and trigonometric expansions (Barron, 1993). Dropout could consistently improve generalization accuracy, and is not only a regularizer for preventing overfitting in neural networks (Liu et al., 2023). The current representations in artificial network might be truly close to the memory storage with thinking and reasoning in the human brain (Oota et al., 2023). One of the recent studies empirically identify the complicated situation in quantifying

the memorization when considering the training data duplication (Carlini et al., 2023). The empirical studies verify that a well-trained DNN usually encodes sparse, transferable, and discriminative concepts, which is partially aligned with human intuition (Li & Zhang, 2023).

Reconsidering the **generalization** utilizes the experimental framework in the pursuit of the theoretical foundations of neural network (Zhang et al., 2021), e.g., the implicit bias of gradient descent on the linearly separable data (Soudry et al., 2018; Frei et al., 2023), the potential benefits of overparametrization with the well-specified neural networks (Hastie et al., 2022). the generalization ability makes the confusion on the unseen data for the noisy and iterative learning algorithms (Dong et al., 2023b).

**Transformer** and the variants are universally the popular architecture in the practices, and the applications are far more fruitful than the original purpose on the task on machine translation (Clark et al., 2019; Sun et al., 2020), including natural language processing (Lin et al., 2022), computer vision (Kashefi et al., 2023), medical image diagnosis (Zhu & Wang, 2023), times series (Wen et al., 2023), and etc. The transformer based models are one of the great triumphs in the recent renaissance of the research on neural network. The **in context learning** capability emerges with the extreme volume of the parameters from the large-scale language models (Dong et al., 2023a; Bai et al., 2023; Dong et al., 2023b) . There is the inappropriate drawback of the multi-stage training with the occurrence of catastrophic forgetting of prior knowledge (Dong et al., 2024). The accumulated knowledge rarely considers the emotional feelings and personalized attitudes with the scope in the philosophy foundations (Asai et al., 2020; Wei et al., 2023), and less incorporates the human values in the initial learning procedures and material orientation.

**Hallucinations** and harmful content are considered as the undesirable artifact of the large scale language model. The generative model provides the bad response in the certain scenarios of the prompting containing illegal actions, offensive language abusing, personal privacy leakage, cybersecurity threatening, unqualified professional advice (Ji et al., 2023). The numerous efforts on the red teaming exhaust the limited resources on fine tuning objectives with the amounts of the model security issues with guardrails on hallucinations and harmful contents(Wei et al., 2023). The persuasive adversarial prompts could increase the jailbreak risk to generate the context without the fully protection on the safety issues(Zeng et al., 2024). The supporting document integration and retrieval augmented generation alleviate the hallucinated content in the response regeneration of LLM-based chatbots (Li et al., 2024). GradSafe accurately detects the jailbreak prompts without necessitating further fine tuning on Llama-2 with safety-critical gradient analysis (Xie

et al., 2024). In some real cases, it is acceptable in rejecting the seemingly toxic prompts to prevent malicious output with the side effect in sacrificing answering the innocuous prompts (Cui et al., 2024). DeepSeek utilizes rule-based rewards in mathematics, coding and logical reasoning domain, and captures human preferences in mitigating the potential risks, biases, or harmful content in the generation process (DeepSeek-AI et al., 2025).

**Chain of thought** reasoning is the intermediate steps with the multiple reasoning paths and self-evaluating choices, and the recent survey refers to (Chu et al., 2024), e.g., looking ahead or backtracking in the strategic decisions with the conscious mode (Yao et al., 2023; Prystawski & Goodman, 2023), solving complex mathematics and sequential decision-making problems from the human experiences (Feng et al., 2023).

**Reinvigorated neuroscience** research recently demonstrates on memory retrieval in human brains with the shallow evidence (Josselyn & Tonegawa, 2020). The episodic memory and semantic memory are the structuralized category on the internal mechanism in the brain (Budson, 2009). In the recent **connectome** experiments, almost the majority of the neurons therein are extensively connected, but only few neurons are functioning well in the internal competition on the specific tasks (Ripoll-Sánchez et al., 2023; Randi et al., 2023; Winding et al., 2023).

The **eXplainable Artificial Intelligence** is attractive for unlocking the inner mechanism of the neural network (Buchholz et al., 2023; Lorini, 2023; Yu et al., 2023b), and the standardized evaluation is not complete with the decent coverage and the comparison transparency (Le et al., 2023; Delaney et al., 2023). Incremental XAI to automatically partition explanations for general and atypical instances to help users read and remember more faithful explanations (Bo et al., 2024). The hybrid fusion is empowering the domain experts and data-centric explanations collaborating in the interactive systems (Bhattacharya et al., 2024). The interpretability and analysis research is pursuing the deep understanding of inner workings of large language model (Mosbach et al., 2024). The explainable AI is still in the infancy with the earlier prototype on the evaluation and benchmark.

In the recent trials on the interpretation of the network mechanism, **causal model** connects the observed variables with the entire system in the interventionist conditionals. The causality initially brings the methods and results from the accumulation in machine learning and philosophy. The theoretic objective of the casual discovery is finding the underlying causal relationships among the observed variables in the earlier trials (Park et al., 2023), e.g., the interdisciplinary practices of the individual treatment effects (Alaa et al., 2023; Imbens & Rubin, 2015; Gunsilius, 2023), the

system of the structural equation modeling in the causal reasoning (Lorini, 2023), and the directed acyclic graph to explain the causality in the contemporary machine learning (Squires & Uhler, 2023).

**Logic rules** are the building blocks in reasoning with the symbolic models in the domination era of the expert system on the complex system (Bozorgi et al., 2020; Yu et al., 2023a), and the authors in the field of fuzzy logic recently revitalize the interest for the explainable artificial intelligence challenge with the interpretability and accuracy trade-off (Alonso Moral et al., 2021). The symbolic rules follow the mathematical spirit in the previous generation of expert system. The fuzzy rule-based models with the ensemble strategy suggest the nonlinear characteristics and substantial interpretability (Hu et al., 2019). One of the limitation is that fuzzy logic rule requires the tremendous efforts with the expert knowledge in the mature accumulation on the specific domain (Alonso Moral et al., 2021; Yu et al., 2023a; Baldoni et al., 2018).

## 3. Complexity on Neural Network

The neural network is superior than the linear model and the other basic machine learning algorithms with the benchmark comparison and the concrete examples from academic papers and industrial competitions. The neural network provides the new representation of the diverse world, and the form is more likely to align the complex relations among all the possible hierarchy in the reality. The neural network is more superior than the regression based models because the neurons are more flexibility with the individual heterogeneity in competition and interdependence.

[Interpretability Hypothesis] The neural network cannot (at least has the close to zero probability) be transparently explained by the series of linear equations. The linear equation is one of the specialized forms of the computation in neural network. The neural network is powerful and efficient with the properly designed architecture in solving the amounts of well-studied problems in practices. The neural network is likely to be the more advanced system with the empirical verification and compelling performance than the linear equation.

The neural network advances the performance accuracy because the things are reciprocally flexible in the connected layers instead of the linear loop or the sequential steps summarized in the logical rules. The neurons are probably the elementary elements carrying more effective information and multiple relations in the neural networks than the variables doing in the linear models and the advanced functions. The phenomenon, incident, and circumstance are much more complex than the single variable described as the predictors and predictions in linear regression models. The obvious

limitation for linear model and function is incorporating the compound relations to simulate in the real circumstances with the hidden status.

The linear model is generally simplifying the real scenarios into the arbitrary additive components with the unexpected error and system bias. The complexity is not always the priority on the regression-based procedures in decomposing and perceiving the world. The theoretical pursuit is not sudden appearing on the dilemma of interpretability and performance in machine learning, and the theoretical simplicity originates back into the elegance of the modern statistics and earlier econometrics (Aydogan et al., 2023). The low dimensional analysis emphasizes on the correlations and interpretability in the linear equations ideally with intended approximation, but the high dimensional data has already exhausted the initial idea into the unpaved mess.

The interpretability hypothesis replaces the linearity assumption with the new paradigm to review the the strong assumptions in earlier science community. The interpretability hypothesis stands as the opposite side of perceiving the symbolizing world in the decomposition of elementary components and unexpected parts. The internal connections are complicated with the individual heterogeneity in neural network and transformer based models, and the linear model is the simple model to capture the inaccurate projection of the compound relations in the equations and functions among the diverse situations in reality.

## 4. Neuron Flexibility and Abstract Presentation

The complex interdependence and neuron flexibility synergistically make the interpretation difficulty for the neural network and transformer based model. The representation is abstract with the weights and the parameters into the intrinsic topological structure through the training process. The neuron activity is flexible among the neighbors and the arbitrary layer in the system. The interdependence is too complex to enumerate the neuron relationships in sequence among the internal cooperation of the neural network. One earlier example is the interpretation the prominent performance of network with the dropout structure, and the claiming is not proper for dropout in the literature.

**Corollary 4.1** (Neuron Flexibility)**.** *The neurons have at least the two types of connections in the system, the standing status but silently in some situation, and the key status with the higher relevance in the activating forms.*

The corollary is the assumption for categorizing the connection types of flexible neuron, and the more types of the neuron connections would dramatically improve the flexibility of the internal activity with the higher level. The things are all universally connected in the world, but not all vari-

*Table 1.* Characteristics on neural network and linear models

| Algorithms | Linear/additive model | Neural network/Transformer-based |
| --- | --- | --- |
| System Flexibility Complexity | Manual spell with previous knowledge Limited capacity in equation and bias projected Simplicity with the specialized form | End to end framework with internal compensation Diverse relationship among real world No-linearity in the neuron activity |

ables are equivalently functioning straight in equalizing the relationship with all the situation in the full connection. The full connected neurons restrict the complexity of the internal structure into the status with the less cooperation. The existence of the standing status vibrantly enriches the networks with the possibilities of the randomness, and strengthens the key player with the multiple potential exempting in the network.

The representation is highly abstract to be beyond the initial purpose of the neural network in the initial purpose and generalized capability. The internal structure only evolves and derives in the process of obtaining the particular neural network, and the neuron activity is not always relevant with the mechanism in reasoning and inference on real problems. The standing status facilitates the system with the extra freedom in solving the arbitrary tasks.

## 5. Common State and Reverse Availability

[Universality Hypothesis]The state is the combination of the current representation and the feeding data under the condition of the initial structure and training recipe. The common state has the highest performance, and the inner structure is feasible to mutate into the universal form in any layers (network depth) and neurons quantity (network width).

The common state distinctively outperforms from the candidates with the adaptation for majority of the tasks on the inference and prediction. The common state is outstanding with the extreme flexibility on the multiple reasoning objectives, and the required procedures and efforts are relatively the minimum in the competition with the obtained states. The common state has the universal capability of the accuracy and efficiency in general, but it is currently unclear on the mechanism to identify the common state with the mutation from the multiple candidates with the precise measurement over the training procedures.

The common state is the same form generalized for all the underlying neural network without considering the proper scale of parameters in the internal structure. All the neural networks intrinsically share the common state with the equivalent representation as the universal form. The common state is the essence as the intermediate state with the stemming capacity on producing the arbitrary representation with the weights and the layers in the manipulative selection.

The technique and the solution remain an open question to manipulate the state with steering the representation to differentiate the internal activity with the progressive transfer learning.

**Corollary 5.1** (Reverse Availability). *The common states (or the ultimate network) and the neighboring states are gradually consolidated by appropriately aggregating the multiple small networks with the initial separate purpose.*

The common state at least has one approximating equivalence in connecting with the smaller scaled network. The common state efficiently leverages neuron flexibility with the feeding data and the universal representation. The current achievement of LLM seems not the unique solution on the inner structure for the neural network with the emergent capability beyond our expectation. From the universality hypothesis and the reverse availability, there is the limited evidence on the evolving process from the candidate state to the common state.

Aggregating the multiple individual networks with the proper topological connection would be one of the alternative ways for obtaining the common state from the large network. The neural network in the large scale is probably emerging in the complex representation with the general reasoning and perceived cognition for the universe, and the neural network in the small size is the building block for arbitrary selecting the representation of the common state. The accumulation process simultaneously absorbs the advantage and shortcoming from the smaller potential networks, and optimizes in the computation efficiency with the model capability and reasoning ability in the general purpose.

It is more meaningful and valuable to prioritize in identifying the common states technically and purposefully than theoretically unblocking the suddenly acquired ability in the parameter scale under the current wave of large language models. It is feasible to incorporate the multiple trained models with the delicate design in the alternative network structure to obtain the common state of the general representation with the proper manipulation and transformational forms.

## 6. Shadowing Deficiency from Human

[Deficiency Hypothesis] The imperfection and inefficiency

Table 2. The representation and data together formed the state in the status

| Model Status | Common State | Candidate State |
|---|---|---|
| Property | Accuracy and efficiency | Semi accurate in the performance |
| Utility | Universal form to adapt | Initial structure in the design |
| Solution | Active manipulation (unknown) | Selected from the training process |

of neural network are owing to the imitation of neuron activity in human brains, as which bias and errors diversely appear in redundancy. The neural network is partly echoing the property and deficiency of the human brain, whereas the incomplete observation leads to the unclear description for the internal activity of the brain from neuroscience. The reoccurrence of the imperfection is natural in behavior with the repetition and redundancy in the internal activity. One of the cognitive example is that most people still have the chance to make mistakes for the simple repetitive tasks without sufficient practice and training in the guidance, and many more difficult jobs deserve to learn and study in the accumulation of human intelligence.

The neural network and transformer based model do not follow linear strategy and logic rules discovered from symbolism era and previous world. The functions and logic rules are the concepts inheriting from the mathematics and symbolism models, and the data and representation become the new form in the human interaction and competition without control. The common state and the candidates are more suitable to foresee the world with the data and representation, and it is shifting the finite form of the functions into the infinite structure with the complexity and flexibility.

Lacking human sense is not problematic in individual personality for the prototype of general intelligence as the reliable helper and trustful advisor. There would be the open debate on the argument that the state simultaneously acquires the emotion and consciousness as the emerged capability with the complexity of the neuron activities. The continuous efforts inevitably follow the Turing's idea on the machine intelligence in pioneer, the earlier assumption is that the machine mechanically mimics the human reasoning with the free will towards the breakthrough of the potential artificial general intelligence (Turing, 1950).

The co-occurrence of the emerged ability and unexpected hallucination is the human like deficiency for the model and machine. The emergent capability simultaneously occurs with the side effect of hallucinations in the large scaled model. The ongoing debate remains open for the discussion on the inner mechanism of the phenomenon, and the empirical study is much more challenging to examine the hypothesis and conjecture jointly on the large language model beyond the normal level of the human intelligence.

The foundational hypothesis is potentially opening the new direction for all the scientists and communities to rethink the failure and incompleteness in contemporary mathematics.

## 7. Discussion and Suggestion

The network hypothesis is not directly concentrating on the structure design and model training in the practice, but the new paradigm is helpful and promising for understanding the unrevealed secrets of neural network in imitating the human reasoning and perceptions.

### 7.1. Ensemble

The model ensemble is the special case in approximating the neural network for the real problems. The bagging or boosting schemes seem like the connecting neurons in the network, and the selected models are the prior knowledge extracted from the specific problems. The last averaging pooling step tolerances the differences on the output results with the intended ambiguity among the bagging models. The weight allocation approximates to simplify the neuron activity in the network.

The bagging and boosting schemes are more efficient than the regression based models because the system is balancing the heterogeneity with the flexibility as the specialized advantage in combining the individual models. The model ensemble still has the worse performance in results than the neural network because the individual model in ensemble strategy still follows the straight additive strategy incompletely describing the problem in real.

### 7.2. Dropout

The explanation is ambiguity on the internal mechanism of the network with dropout in the previous claim. The interpretation is neglecting the richness in neuron status and the assembling creativity unlocked from the network structure with dropout. The connected neurons could have the diverse variety of the status on the entirely full connection in the network (Corollary 1, Section 4), and the neurons are not simultaneously functioning as the equalized participants in all the situation of the problem. Some neurons might have the specialized reasoning role in some situations, and might be only the standing or supporting role in the form with the path transition in the complex system.

The claim is inappropriate on avoid overfitting as their pro-

*Table 3.* Differences on connectionism and symbolism

| Character | Connectionism | Symbolism |
|---|---|---|
| Formation | LLM in parallel computation | Program running on computing machine |
| Internal | State with data and representation | Functions and logic rules in mathematics |
| Control | Resource competition in business | Human made in the finite forms |

found instructions on their initial proposal (Srivastava et al., 2014), and the value of the dropout network is highly undervalued and still less discussed in the deep learning community. The dropout almost plays an important role consolidating the internal complex on recognizing the handwritten digits in MNIST dataset,[1] and the intuition lacks the technical prove with the structure in many task-oriented networks originated in computer vision (Kong et al., 2022). Dropout is not only the operating component for the internal system to avoid overfitting with the training data (Liu et al., 2023), but the enormous silence is refreshing the new potential state to reconstruct the internal activity in the complexity and flexibility.

### 7.3. Breakable Rules

The representation surpasses the invalid pursuit in incorporating the overlapping or intermediate status into the finite form of the rules for the state of the neural network. The neural network is complex and flexible to alternatively understand the universe precisely from the inner structure other than the simulated equalization and separated errors. The neural network and transformer based models provide the descent solution considering the breakable rules and weak rules in connectionism. The incompatibility is the failure of the previous methods on the overlapping or intermediate status with the mathematics foundation.

### 7.4. Duplication

The recent LLMs are beneficiary form removing the duplication context with the hundreds of gigabytes of training data, but the mechanism is theoretically unclear for the improvement in initialing the same structure with emerged capacity. Intuitively, the model is incapable of differentiating the multiple sources of the training material in the states with the representation of the illicit opinions in the crawled online data and the licensed research data. The obtained representation might be in the disorder with the duplicate and similarity in the learning context as the response from the underlying internal reaction to perceive the observed world in the human brain. The limitation still remains the challenge to entirely remove the numerous false material in training the new model again from the bottom with the same complexity and equivalent capacity.

---

[1] http://yann.lecun.com/exdb/mnist/

### 7.5. Linguistics

People know, recognize and perceive the things and reoccurrence with the natural language. The neural network aligns the state from the natural language, and relies on the interdependence from the elementary blocks of stem, phrase, and passage in the corpus. The obtained model reflects the real world in the mirror with the common wording and variant phrasing in the natural language, and the representation do not only capture and store the numerous information from the amounts of training data in the reordered sequence. The transformer based model suggest the unknown mechanism is paralleling with the conventional linguistic characteristics on word, sentence and chapter. Previous computational modeling neglects the unknown mechanism connecting the world with the natural language.

### 7.6. Memory

The interpretability research has the opportunity to advance the position to understand the formation of the memory and internal activity of neural network. The neuroscience and connectome experiments do not contribute enough to guide the future direction of the neural network and transformer based model. The brain and neural network might operate in the unknown mechanism to avoid seeking the memory storage technically in the hierarchy structure. Categorizing with episodic memory and semantic memory is the theoretical conjecture in memory retrieval rather than empirical observation from human brain. The memory retrieval and storage is only the external process writing results in the software defined computer. The large language models do not only memorize the data available in pretraining and fine tuning. The state is more flexible and resilient with internalizing the feeding data and representation in the infinite form.

### 7.7. Innovation

The community should encourage the new ideas, open technics, and transparency on the academic research in the corroborative industrial commercialization environment, and restrict the unexpected use with the responsible framework to balance the social welfare and the illicit risks of the potential emergence of the general artificial intelligence.

The research and business instances would become concrete

and solid continuously with the complete information disclosure on data, tricks and recipe to reproduce the current models for fulfilling the incompleteness and deficiency of the large language models (e.g., the spatial and temporal resolution, computation efficiency, hallucinations, etc.). Some model owners now partly take the relatively conservative attitude as the developer responsibility to alleviate the harmful scenarios in the frontier technology application. The regulators have the dilemma on the safety prevention for the public release, and the behavior surveillance is probably easing the rapid development of current LLMs. The close strategy is the premature closure hampering the research advancement and potential innovation on the modeling.

## 8. Limitation

The network hypothesis is not formulating the complete theory with the immediate actions to ruin the interpretability research, but clarifying the reasons and shortcomings of the previous incompleteness in explainable AI. It is a starting point for further discussion on internal structure of neural network and large language models. The established hypothesis and corollary pave the avenues for validation and iterative refinement with technical details to interpretate neural network and large language models in the future peer contribution.

## 9. Alternative Views

The linear models are the special case of the neural network, and the simplified structure makes the realtionship of the variables in the manner of the simlicity and explainability. The appromixation into the linear equation is not harmful in simplifying the complex structure of the neural networks in the real problems.

The undesirable answers are naturally borrowing from the original training material, instead of the generative responses in the loop of the reasoning inference. The large langue models produce the hallucinations because of the wrong structure settings in the pretraining and opaqueness in the training materials.

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
