# OpenReview forum: "Position: Explainable AI Is Misleading Paradigm Shifting of Neural Network"
_ICML.cc/2026/Position_Paper_Track — Submitted to ICML 2026 Position Paper Track_

### Official Review · Reviewer_7C9r · 2026-03-08

**Significance:** 2
**Argument Clarity:** 2
**Rating:** 3
**Confidence:** 3

**Questions:**

1.	If NNs possess an "infinite structure" that invalidates current theoretical frameworks, how would you formally define this boundary? Is there a preliminary mathematical framework you can provide?
2.	Regarding the "Reverse Availability" corollary: Do you have any empirical evidence or theoretical proofs demonstrating how this proposed topological aggregation differs in efficiency or reasoning capability from standard model ensembles?
3.	At what specific complexity threshold or evolutionary stage do you posit that a system driven by gradient-based optimization transitions from "mathematical computation" into your proposed "new scientific paradigm"?
4.	While the concept of "Standing status" neurons is conceptually interesting, how do you propose to empirically test or verify their role in providing "extra freedom" during reasoning in future work?

**Alternative Views Section:**

Yes

**Compliance With Llm Reviewing Policy A Conservative:**

Affirmed.

**Discussion Potential:**

2

**Final Justification:**

After reading the other reviewers' comments and the authors' rebuttal, I do not think the rebuttal has fully addressed the main criticisms of the paper. As a final justification, I maintain my original evaluation and still believe that the work does not yet meet the quality standard expected for an ICML position paper.

**Paper Summary:**

This position paper argues that neural networks (NNs) represent a new scientific paradigm that fundamentally transcends traditional mathematics and rule-based logic. The authors propose four main hypotheses (Network, Interpretability, Universality, and Deficiency) to contend that NNs cannot be adequately reduced to linear equations, and that phenomena like hallucinations are inherent byproducts of mimicking human cognition. Additionally, the paper introduces a "Reverse Availability" corollary, suggesting that a universal "Common State" can be achieved by topologically aggregating smaller networks.

**Position:**

Yes

**Position In Title:**

Yes

**Related Work:**

3

**Strengths And Weaknesses:**

Strengths:

1. The discussion on "Neuron Flexibility" (specifically distinguishing between standing and key statuses) offers a useful conceptual lens for examining polysemantic neurons.
2. Framing Dropout as a mechanism for "system state refresh" rather than a mere regularizer, alongside linking emergent abilities with hallucinations, presents an interesting theoretical perspective on LLM behaviors.

Weaknesses:

1. Although the authors claim that neural networks shift into an “infinite structure,” the manuscript does not provide a rigorous mathematical formalization of this concept (for example, from the perspective of set theory or nonlinear dynamical systems). As a result, it is difficult to precisely understand what this notion entails.

2. The key arguments related to the “Common State” and “Reverse Availability” are currently not supported by sufficient empirical evidence. In particular, the claim that aggregating small networks can lead to the emergence of such states would be more convincing if accompanied by ablation studies, simulations, or other experimental validation.

**Support:**

2

---

> ### Author Rebuttal · Authors · 2026-03-27
>
> 1. The infinite structure describes the neural network with complex possibility and extreme flexibility, and the internal weights are almost changing all the way in every epoch of the model training. The unstability of the performance in generalization is supportive for the infinite property with the unexpected catastrophic forgetting in many scenarios of the practices. Moreover, the neural network achieves the great success with the neuron flexibility to generate the complex possibility to perceive the enormous things and diverse relations in understanding the real world. All the neural network follows the hypothesis in the position if none of them is verified to be the false statement. As the position is questioning some foundation of the science, the mathematics and logic rules are not still the valuable tools any more to provide some preliminary thinking and complete analysis for the theoretical problems of the neural network boundary. The position paper is alternatively utilizing the powers and semantic meanings in the natural language instead of constructing the equations with the limitation of the one way simplification of the equalizations.
>
> 2. The reverse availability is the corollary from the universality hypothesis of the common state, and the common state is the perfect abstraction to distinct the uniqueness of the training candidates of the neural network in the theoretical imagination. The hypothesis is additionally helpful in understanding the essence of the invariable states of the network training (if the entire structure can remove or add some new blocks of the networks as the new ingredients), even we do not have the solution to select the ideal one from the candidate states now, like we almost have no idea for assessing the same model with different epoches in the training process. In the unverified hypothesis, the common state is universal to generalize all the neural networks with the prominent abilities with the efficiency in the universe. Model ensembles always provide the better results than any individual models with the specific strengths and weakness, and the common state defines the only feasible structure equivalent to any state of neural network.
>
> 3. There is no swift transition with the changing threshold, and the current neural network itself represents for the new paradigm when their compelling property is already not subordinated to the mathematics, rules, and logics any more. The statement and hypothesis are just the clarification on the difference between the previous science paradigm and new shaping trends in perceiving the world. The mathematical computation is the process that we are building the neural network and large language models, and the new scientific paradigm is getting rid of the science foundations in exploiting the equations to stimulate the things in the universe. The capability of the neural network is far more beyond the dilemma of the equations and logic rules that we can understand now in the science development. The new paradigm follows the doctrine and essence of the neural network whether we would have the wrong opinions on the current achievement and potential emergent of neural network. The hypothesis is standing as the earlier attempts to rethink the situation with the decent position in the alternative perspectives, which the efforts are worthwhile in spurring the thorough revolution in lagging with the current success of the construction in practices.
>
> 4. The neuron flexibility, dropout structure (more tricks in recent modeling), and complex possibility collectively constitute the general landscape of the neural network. The neuron flexibility is not the new concepts for understanding the polysemantic neurons in the recent works (some prior works illustrate the internal weighting of the response to the multiple objects), and the standing status and key status are the extreme simplification of identifying the multiple status of the neurons working and reasoning within the internal structure of the network. The extra freedom is first bringing into the system when we initially found the valuable structure with dropout neurons, but no paper notices that the growing complexity is due to adding the dropout in network, even not in the intuitive claim in community. The empirical work is not difficult, but requires formally setting the linear or loop sequences aside in simplifying the flexible neurons to emerge with the capabilities in the practices. The network complexity is potentially hauling the identification of polysemantic neurons with the expanding layers and structures in the large language models, and the simplified model is struggling to obtain the complexity in the same level. The new paradigm is forcing us to realize and respect the complexity of the neural network and large language models, and rewrite many foundational ideas for more unseen miracles.

---

> > ### Author Rebuttal · Reviewer_7C9r · 2026-04-01
> >
> > After carefully reading the authors’ rebuttal together with the comments from the other reviewers, I acknowledge that the authors have clarified some of the concerns I raised. However, when considered alongside the key issues identified by several other reviewers, I do not think the rebuttal has fully addressed the main criticisms of the paper. Taking all of these factors into account, I maintain my original evaluation and still believe that the work does not yet meet the quality standard expected for an ICML position paper.

---

### Official Review · Reviewer_TvGy · 2026-03-12

**Significance:** 1
**Argument Clarity:** 1
**Rating:** 1
**Confidence:** 5

**Questions:**

What generative AI was used to write this paper ?

**Alternative Views Section:**

Yes

**Compliance With Llm Reviewing Policy A Conservative:**

Affirmed.

**Discussion Potential:**

1

**Paper Summary:**

It is not clear what the contributions of this paper are, and the position it advocates.

**Position:**

Yes

**Position In Title:**

Yes

**Related Work:**

1

**Strengths And Weaknesses:**

Strengths:
None


Weaknesses:
The paper is unreadable, with too many sentences that cannot be apprehended, and too many errors.

The paper was probably written by a generative AI, leading to sloppier paper. A good representative example is the abstract, with “The position paper argues that the neural network is shaping the new paradigm of science, and neural networks are generally superior to the rule, logic, and mathematics. The network hypothesis emphasizes the superiority of the neural network: …”

This goes also for the stated position, which is not the same in every part of the paper. For instance, the position highlighted in the title is not the same in the abstract, and the rest of the paper does not provide a clear definition of the position of the paper.

Thus, what is the stated position? And therefore, it is not well supported and it is not clear what its relevance and significance are to the ICML community.

The paper seems to be written as a survey, but it misses many aspects and references. It fails to be a comprehensive survey.

The Alternative Views section is tiny and does not provide alternative views of any position.

**Support:**

1

---

> ### Author Rebuttal · Authors · 2026-03-28
>
> The position in the abstract and introduction is the decent explanation on the claim in the title, and both the presentation is clear and precise enough to claim the views on alternative perspective to understand the complicated situation with the superiority of model performance among the invalid foundation of the theory on neural network. The community is neglecting the influences of the new shaping paradigm bringing from the neural network and large language model, and the hypothesis is weakening the claim of the position with the limited evidences and mathematic proofs now (it is relatively the conservative way to present the question and thinking above the previous opinions). The verification and discussion are awaiting the contribution in the future studies among the pioneering peers with the community. The main hypotheses (Interpretability, Universality, and Deficiency) touch the fundamental problem of the compelling property of the neural network in the pursuit of the unveiling the secrets, and the attempts are effective in explaining the dropout structure and ensemble in machine learning. It is still valuable to rethink the current paradigm on the science and technology with the challenge and potential harmful scenarios of the emergent of the available models.
>
> There is the contradiction between generative AI and making mistakes as in your comment with the bias, and the paper is absolutely not relying on any input with the large language models. We still insist on thinking and writing with our own privilege in the scientific research, even we have the latest knowledge on the emergent capabilities of the large language models.

---

> > ### Author Rebuttal · Reviewer_TvGy · 2026-04-04
> >
> > I would like to thank the authors for the rebuttal. However, the rebuttal does not cover the major raised issues in my review. A specific example is that I still do not know specifically and concretely what is the position of the paper.
> > Moreover, the authors do not reply on all issues raised in my review, such as on the Alternative Views section.

---

### Official Review · Reviewer_6xoS · 2026-03-13

**Significance:** 1
**Argument Clarity:** 1
**Rating:** 1
**Confidence:** 5

**Questions:**

N/A

**Alternative Views Section:**

No

**Compliance With Llm Reviewing Policy A Conservative:**

Affirmed.

**Discussion Potential:**

1

**Final Justification:**

This paper is completely unreadable, so I strongly recommend rejection. There is no rebuttal to my review.

**Paper Summary:**

The position of this paper, as I understand it, is that neural networks are superior to previous rules-based algorithms and mathematical approaches.

**Position:**

No

**Position In Title:**

No

**Related Work:**

1

**Strengths And Weaknesses:**

This paper needs a major rewrite to make it understandable. Grammar issues, spelling issues, and vague, unclear, or imprecise language make it very difficult to understand. I have difficulty understanding large portions of this paper. In particular, this makes it difficult to understand how the author is supporting their position and what the alternative views are. I also believe that the position of this paper, as I understand it (“neural networks are superior to previous rules-based algorithms and mathematical approaches”) does not meet the criteria for ICML position papers.

**Support:**

1

---

### Decision · Program_Chairs · 2026-04-30

**Decision:**

Reject

**Comment:**

This position paper argues that neural networks constitute a new scientific paradigm that transcends traditional mathematics and rule-based logic. The authors introduce several hypotheses, including "infinite structure," "neuron flexibility," and "Reverse Availability," to characterize the behavior of modern models.

While reviewers noted some conceptually interesting perspectives regarding polysemantic neurons and dropout as a system-state refresh, the submission falls well below the minimum standards for publication. Reviewers 6x0S and TvGy emphasized that the manuscript is largely unreadable due to pervasive grammatical errors and imprecise language, which obscures the intended contributions and makes the core position impossible to evaluate. Additionally, reviewer 7C9r observed that the central claims lack the necessary mathematical formalization and empirical evidence, particularly regarding the proposed "Common State." The authors' rebuttal failed to resolve these fundamental issues or provide clarity on the paper's central thesis.